# Bifunctional Peptidomimetic G Protein-Biased Mu-Opioid Receptor Agonist and Neuropeptide FF Receptor Antagonist KGFF09 Shows Efficacy in Visceral Pain without Rewarding Effects after Subcutaneous Administration in Mice

**DOI:** 10.3390/molecules27248785

**Published:** 2022-12-11

**Authors:** Maria Dumitrascuta, Charlotte Martin, Steven Ballet, Mariana Spetea

**Affiliations:** 1Department of Pharmaceutical Chemistry, Institute of Pharmacy and Center for Molecular Biosciences Innsbruck (CMBI), University of Innsbruck, Innrain 80-82, 6020 Innsbruck, Austria; 2Departments of Chemistry and Bioengineering Sciences, Vrije Universiteit Brussel, Pleinlaan 2, B-1050 Brussels, Belgium

**Keywords:** multifunctional ligands, mu-opioid receptor, neuropeptide FF receptors, G protein-biased-opioid agonist, pain, opioid analgesia, opioid epidemic, rewarding effect, sedation

## Abstract

There is still an unmet clinical need to develop new pharmaceuticals for effective and safe pain management. Current pharmacotherapy offers unsatisfactory solutions due to serious side effects related to the chronic use of opioid drugs. Prescription opioids produce analgesia through activation of the mu-opioid receptor (MOR) and are major contributors to the current opioid crisis. Multifunctional ligands possessing activity at more than one receptor represent a prominent therapeutic approach for the treatment of pain with fewer adverse effects. We recently reported on the design of a bifunctional MOR agonist/neuropeptide FF receptor (NPFFR) antagonist peptididomimetic, **KGFF09** (H-Dmt-DArg-Aba-βAla-Bpa-Phe-NH_2_), and its antinociceptive effects after subcutaneous (s.c.) administration in acute and persistent pain in mice with reduced propensity for unwanted side effects. In this study, we further investigated the antinociceptive properties of **KGFF09** in a mouse model of visceral pain after s.c. administration and the potential for opioid-related liabilities of rewarding and sedation/locomotor dysfunction following chronic treatment. **KGFF09** produced a significant dose-dependent inhibition of the writhing behavior in the acetic acid-induced writhing assay with increased potency when compared to morphine. We also demonstrated the absence of harmful effects caused by typical MOR agonists, i.e., rewarding effects (conditioned-place preference test) and sedation/locomotor impairment (open-field test), at a dose shown to be highly effective in inhibiting pain behavior. Consequently, **KGFF09** displayed a favorable benefit/side effect ratio regarding these opioid-related side effects compared to conventional opioid analgesics, such as morphine, underlining the development of dual MOR agonists/NPFFR antagonists as improved treatments for various pain conditions.

## 1. Introduction

Despite the increased prevalence of opioid misuse and abuse over the past years, opioids continue to remain the most important class of analgesic drugs for the treatment of moderate-to-severe pain [1,2,3]. However, opioid safety, particularly upon prolonged use, is dramatically reduced because of numerous and severe side effects, including constipation, respiratory depression, sedation, tolerance, development of dependence and addiction liability [3]. Prescription opioids specifically produce analgesia through the mu-opioid receptor (MOR), a membrane-bound G protein-coupled receptor (GPCR) [3,4]. Unfortunately, prescription opioids are the major contributors to the current opioid crisis [1]. Therefore, new pharmacological approaches and drug discovery strategies are being explored to mitigate opioid side effects, devoid of abuse liability and with efficacy in various pain conditions. Nowadays, the most targeted chemical and pharmacological strategies include multifunctional ligands, G protein-biased agonists and peripherally restricted opioids [5,6,7,8,9,10].

Regarding multitarget drugs, the field of medicinal chemistry has seen an extensive effort toward discovery of more efficient and safer therapeutics for human disorders using such an approach [11,12]. The concept of multitarget pharmacology/polypharmacology is also applied in the opioid research as a valid strategy to develop new analgesics with improved efficacy and reduced side effects [10,13,14,15,16,17]. Preclinical investigations by us and others reported diverse multifunctional ligands targeting opioid/opioid and opioid/non-opioid receptors as potential new pain therapeutics [13,14,15,16,17,18,19,20,21].

We have recently reported on a bifunctional peptide-based hybrid, **KGFF09**, which combines a MOR pharmacophore and a neuropeptide FF receptor (NPFFR) pharmacophore (Figure 1) [19]. Additionally, **KGFF09** was described as the first dual ligand acting as a G protein-biased MOR agonist and NPFFR antagonist in vitro (Appendix A). Although **KGFF09** showed agonist activity at the delta opioid receptor (DOR), its binding affinity is 77-fold lower than at the MOR. **KGFF09** also interacts with the kappa opioid (KOR) and nociceptin (NOP) receptors, but with an antagonist profile at both receptor types. Behavioral studies showed **KGFF09** to be very effective with long-lasting antinociceptive effects in mouse models of acute pain (i.e., warm water tail-withdrawal test and radiant tail-flick test) and persistent inflammatory pain (complete Freund adjuvant (CFA)-induced hyperalgesia) after subcutaneous (s.c.) administration [19] (Appendix A). Moreover, **KGFF09** presented reduced opioid-induced side effects, including respiratory depression, hyperalgesia, tolerance and withdrawal syndrome at effective antinociceptive doses [19] (Appendix A). Because of this noteworthy pharmacological profile, further investigations of **KGFF09′**s pharmacology were warranted. The present study was undertaken to evaluate antinociceptive activity of **KGFF09** in a mouse model of visceral pain after s.c. administration and to assess potential opioid liabilities for rewarding and locomotor dysfunction following chronic treatment.

## 2. Results and Discussion

### 2.1. In Vitro Pharmacological Properties of KGFF09 as a Highly Potent and G Protein-Biased MOR Agonist

Previous data on the in vitro functional profile of **KGFF09** as a potent and full MOR agonist were reported in the cyclic adenosine monophosphate (cAMP) accumulation assay using human embryonic kidney (HEK293) cells the stably co-expressed the cAMP Glosensor-20F and the human MOR [19] (Appendix A). In the current study, we assessed the in vitro activity of **KGFF09** at the human MOR using the guanosine 5′-*O*-(3-[^35^S]thio)triphosphate ([^35^S]GTPγS) binding assay with membranes from CHO cells that stably expressed the human MOR, as described previously [20]. Agonist potency (ED_50_) and efficacy (E_max_) values are shown in Table 1. Stimulation of [^35^S]GTPγS binding induced by **KGFF09** was compared to the effect of the prototypical reference full MOR agonist [D-Ala^2^,*N*-Me-Phe^4^,Gly-ol^5^]enkephalin (DAMGO).

As shown in Figure 2, **KGFF09** produced a concentration-dependent increase in the [^35^S]GTPγS binding upon ligand binding to the MOR. It was very potent in inducing G protein activation displaying a four-fold increase in potency compared to DAMGO (Table 1). Furthermore, **KGFF09** acted as a full agonist based on its high efficacy (E_max_ = 92%) in inducing a MOR-mediated [^35^S]GTPγS binding, an observation which is in good agreement with previous findings in the cAMP accumulation assay [19] (Appendix A).

In addition to the G protein activation, another important signaling event following agonist binding to GPCRs, including the MOR, is the β-arrestin2-dependent signaling pathway [6,22,23], which is currently regarded as an opportunity to refine therapeutics for the treatment of human diseases, including pain. In our earlier study [19], we reported on the in vitro functional activity of **KGFF09** in terms of MOR-induced β-arrestin2 translocation using the bioluminescence resonance energy transfer (BRET) β-arrestin-2 recruitment assay with HEK293 cells that stably expressed eYFP-tagged β-arrestin-2 transfected with the plasmid encoding MOR-Rluc8 receptor. In this BRET assay, **KGFF09** showed reduced efficacy (acting as a partial agonist) in inducing β-arrestin2 recruitment (Appendix A), and hence it was classified as a G protein-biased MOR agonist [19].

In the present study, we evaluated the capability of **KGFF09** to induce MOR-mediated β-arrestin2 recruitment using the PathHunter β-arrestin2 assay with U2OS cells co-expressing the human MOR and the enzyme acceptor tagged β-arrestin2 fusion protein (U2OS-hMOR-β-arrestin2 cells) according to a previously described procedure [20]. Concentration-response determinations were carried out in parallel with DAMGO, used as a reference full MOR agonist for the assay. We show that **KGFF09** produced concentration-dependent β-arrestin2 recruitment with a very low potency (EC_50_ = 1264 nM) and reduced efficacy of 57% (of DAMGO) (Figure 2B and Table 1).

To evaluate whether **KGFF09** also displays bias toward the activation of G protein- over β-arrestin2-mediated signaling under the current experimental settings, the [^35^S]GTPγS binding vs. PathHunter β-arrestin2 recruitment assays, we compared its activity profile, that is, potency and efficacy to the human MOR, across the two functional assays. Whereas **KGFF09** acted as a full agonist with high G protein coupling potency, **KGFF09** failed to robustly engage β-arrestin2 recruitment, depicted by the marked 153-fold decrease in potency and low efficacy in promoting MOR-β-arrestin2 interaction (Figure 2C and Table 1). Our current data support previous findings [19] and confirms **KGFF09** to be a G protein-biased MOR agonist in vitro.

### 2.2. Subcutaneous Administration of KGFF09 Produces Potent Antinociception in a Mouse Model of Visceral Pain

We have previously described the potent and long-lasting antinociceptive activity of **KGFF09** in mice after s.c. administration in models of acute thermal nociception (i.e., warm water tail-withdrawal test and radiant tail-flick test) and persistent inflammatory pain (CFA–induced hyperalgesia) [19]. In the mouse tail-flick test, **KGFF09** was shown to be three-fold more potent than morphine [19].

In this study, we investigated the antinociceptive effects of **KGFF09** in a model of visceral pain, i.e., the acetic acid-induced writhing assay, after s.c. administration in mice, as described previously [24]. As shown in Figure 3, KGFF09 produced a dose-dependent inhibition of the writhing behavior in mice. Significant effects were observed at **KGFF09** doses of 0.074 and 0.37 µmol/kg, where inhibition percentages of 65 ± 13% and 95 ± 3%, respectively, from saline-treated mice were calculated. The antinociceptive effective dose, ED_50_ value, of 0.079 µmol/kg (95% confidence intervals 0.033–0.19) was determined. Notably, **KGFF09** was 17-fold more potent than morphine in the writhing assay (ED_50_ = 1.36 µmol/kg or 437 µg/kg) [24]. These results on the potent antinociceptive effects of **KGFF09** in visceral pain following s.c. administration in mice complement previous pharmacological results in acute nociceptive and chronic inflammatory pain [19].

### 2.3. KGFF09 Does Not Induce Rewarding Effects in Mice after s.c. Administration

Previously, we reported on the behavioral profile of **KGFF09** in producing less acute side effects (respiratory depression) and chronic side effects (opioid-induced hyperalgesia analgesic tolerance and withdrawal syndrome) after s.c. administration in mice at antinociceptive doses [19]. The improved profile of **KGFF09** on respiratory depression is accounted to its G protein-biased MOR agonism, whereas the NPFFRs antagonism contributes to the beneficial profile in terms of less chronic adverse effects associated with repeated exposure [19].

In this study, we investigated the propensity of **KGFF09** to induce rewarding effects in mice following s.c. administration. This is particularly important given the current rise in the abuse of prescription opioids, commonly MOR agonists. The MOR is established to play a major role in analgesia, but also on the undesirable opioid side effect of reward, with deletion of the gene encoding MOR (*Oprm1*) completely blocking opioid analgesia and reward behavior in rodents [25]. To this aim, we used the conditioned place preference (CPP) paradigm [26], as described previously [21]. The CPP is a well-established model in mice, where narcotic drugs such as morphine with abuse liabilities produce CPP [27]. The dose of 7.4 µmol/kg of **KGFF09** used in the CPP assay was selected based on our earlier findings with **KGFF09** being highly effective in producing an antinociceptive effect in the tail-flick test (7.4 µmol/kg corresponds to the ED90 dose, and also represents the three-fold antinociceptive ED_50_ dose) and in CFA-induced hyperalgesia (7.4 µmol/kg completely reduced pain behavior of mice to mechanical and thermal stimulation) [19]. For consistency with our previous report [19], we chose this dose of **KGFF09** in the CCP test, as it has also been used in other behavioral studies and was established to cause reduced side effects of respiratory depression, opioid-induced hyperalgesia, analgesic tolerance and withdrawal syndrome [19]. After 4 days of conditioning, during which mice received a daily s.c. administration of the drug or saline, we observed that mice conditioned with **KGFF09** (7.4 µmol/kg) demonstrated no significant CPP response compared to saline-treated animals (Figure 4). In the same test, we reported that mice treated daily with 15.5 µmol/kg of morphine developed a significant place preference compared to the saline group [21]. Our current results highlight the promising profile of **KGFF09** regarding the addiction potential at a dose shown to be highly effective in inhibiting pain behavior.

### 2.4. Chronic Administration of KGFF09 Does Not Affect Spontaneous Locomotor Activity of Mice

Conventional MOR analgesics, such as morphine, oxycodone and fentanyl are known to produce sedation and to reduce locomotor activity, which represent side effects that limit their clinical usefulness [28,29,30]. To further address the behavioral properties of **KGFF09**, the effect on locomotor activity after chronic s.c. administration in mice was assessed using the open-field test, a well-established model for evaluating loss of spontaneous locomotion and sedation [31], performed as described previously [21]. Mice were administered daily, for 4 days, **KGFF09** (7.4 µmol/kg) or saline. The dose of **KGFF09** was the same as in the CPP test, and it was selected as described in Section 2.3.

As shown in Figure 5, **KGFF09** produced no impairment in the locomotor activity as no significant differences in the distance traveled between the **KGFF09**-treated mice and the saline group were observed. In contrast, chronic s.c. treatment of mice with morphine (15.5 µmol/kg) produced significant hyperlocomotion from day 1 to day 4 in the open-field test [21]. Based on this, we reveal the safe profile of **KGFF09** regarding sedation and spontaneous locomotor activity following chronic administration.

## 3. Materials and Methods

### 3.1. Chemicals and Reagents

**KGFF09** was prepared as previously described [19]. Cell culture media and supplements were obtained from Sigma–Aldrich Chemicals (St. Louis, MO, USA). Guanosine 5′-*O*-(3-[^35^S]thio)-triphosphate ([^35^S]GTPγS, 1250 C_i_/mmol) was purchased from PerkinElmer (Boston, MA, USA). Guanosine diphosphate (GDP), GTPγS, DAMGO and 2-[4-(2-hydroxyethyl)piperazin-1-yl]ethanesulfonic acid (HEPES) were obtained from Sigma–Aldrich Chemicals (St. Louis, MO, USA). PathHunter detection reagents were obtained from DiscoveRx (Birmingham, UK). All other chemicals were of analytical grade and obtained from standard commercial sources. Test compounds were prepared as 1 mM stocks in water for in vitro assays or dissolved in physiological 0.9% saline solution for in vivo testing, and further diluted to working concentrations in the appropriate medium.

### 3.2. Cell Cultures

CHO cells that stably expressed the human MOR (CHO-hMOR cell line) were kindly provided by Dr. Lawrence Toll (SRI International, Menlo Park, CA). CHO-hMOR cells were grown in Dulbecco’s Modified Eagle’s Medium (DMEM)/Ham’s F12 culture medium supplemented with 10% fetal bovine serum, 0.1% penicillin/streptomycin, 2 mM L-glutamine and 0.4 mg/mL geneticin (G418). U2OS cells stably co-expressing the human MOR and the enzyme acceptor (EA) tagged β-arrestin2 fusion protein (USOS-βarrestin-hMOR-PathHunter cells) (93-0213C3 from DiscoveRx, Birmingham, UK) were cultured in minimum essential medium (MEM) culture medium supplemented with 10% fetal bovine serum, 0.1% penicillin/streptomycin, 2 mM L-glutamine, 0.5 mg/mL geneticin (G418) and 0.25 mg/mL hygromycin. All cell cultures were maintained at 37 °C in a humidified atmosphere of 95% air and 5% CO_2_.

### 3.3. [^35^S]GTPγS Binding Assay for the Human MOR

Binding of [^35^S]GTPγS to membranes from CHO stably expressing the human MOR was conducted according to the published procedure [20]. Membranes from CHO-hMOR cells were prepared as described previously [20]. Briefly, CHO-hMOR cells grown at confluence were removed from the culture plates by scraping and were homogenized in 50 mM Tris-HCl buffer (pH 7.7) using a Polytron homogenizer, then centrifuged once and washed by an additional centrifugation at 27,000× *g* for 15 min at 4 °C. The final pellet was resuspended in 50 mM Tris-HCl buffer (pH 7.7) and stored at –80 °C until use. The protein content of cell membrane preparations was determined using the method of Bradford with bovine serum albumin as the standard [32]. Cell membranes (5–10 µg) in Buffer A (20 mM HEPES, 10 mM MgCl2 and 100 mM NaCl, pH 7.4) were incubated with 0.05 nM [^35^S]GTPγS, 10 µM GDP and various concentrations of the test compounds in a final volume of 1 mL, for 60 min at 25 °C. Non-specific binding was determined using 10 µM GTPγS, and the basal binding was determined in the absence of the test compound. Samples were filtered over Whatman GF/B glass fiber filters using a Brandel M24R cell harvester (Brandel, Gaithersburg, MD, USA). Radioactivity retained on the filters was counted using liquid scintillation counting with a Beckman Coulter LS6500 (Beckman Coulter Inc., Fullerton, CA, USA). All experiments were performed in duplicate and repeated three times with independently prepared samples.

### 3.4. β-Arrestin2 Recruitment Assay for the Human MOR

The measurement of hMOR stimulated β-arrestin2 recruitment was performed using the PathHunter^®^ β-arrestin2 assay (DiscoveRx, Birmingham, UK) according to the published procedure [20]. U2OS cells that stably co-expressed the human MOR and the enzyme acceptor (EA) tagged β-arrestin2 fusion protein (U2OS-hMOR-βarrestin2 cells) were seeded in cell plating medium into 384-well white plates (Greiner Bio-One, Kremsmünster, Austria) at a density of 5000 cells in 20 μL per well, and maintained at 37 °C for 24 h. After incubation with various concentrations of the test compounds in phosphate buffered saline (PBS) for 90 min at 37 °C, the detection mix was added, and incubation was continued for an additional 60 min at room temperature. Chemiluminescence was measured with the PHERAstar FSX plate reader (BMG Labtech, Germany). All experiments were performed in duplicate and repeated three times with independently prepared samples.

### 3.5. Animals and Drug Administration

Experiments were performed in male CD1 mice (8–10 weeks old, 30–35 g body weight) purchased from Janvier Labs (Le Genest-Saint-Isle, France). All animal care and experimental procedures were in accordance with the ethical guidelines for the animal welfare standards of the European Communities Council Directive (2010/63/EU), and were approved by the Committee of Animal Care of the Austrian Federal Ministry of Science and Research. Mice were group-housed in a temperature-controlled specific pathogen free room with a 12 h light/dark cycle and with free access to food and water. The test compounds were prepared in sterile physiological saline (0.9%). Either the vehicle (saline) or test compounds were administered s.c. in a volume of 10 µL/g body weight. Upon completion of the experiments, mice were euthanized by inhalation of carbon dioxide.

### 3.6. Acetic Acid-Induced Writhing Assay

Writhing was induced in mice by intraperitoneal (i.p.) injection of a 0.6% acetic acid aqueous solution, as described previously [24]. Following a habituation period of 15 min to individual transparent observation chambers, mice were s.c. administered saline (control) or a test drug and after 25 min (5 min prior to testing), each animal received an i.p. injection of acetic acid solution. The number of writhes was counted during a 10 min observation period. Each experimental group included 5–6 animals.

### 3.7. Conditioned Place Preference

The CPP test was conducted in a custom made, three-chamber apparatus (two conditioning compartments separated by a neutral chamber), as described previously [21]. All experiments were performed in an isolated noise-free room illuminated to 60 Lux. The protocol was performed in three phases. (1) The preconditioning phase, during which drug-naive mice were placed in the center compartment and the doors were opened so both test chambers were accessible. The time spent in each chamber was recorded for 15 min. (2) The conditioning phase lasted 4 days, during which the conditioning chambers were closed. In the morning of each conditioning day, mice received saline and were placed individually for 30 min in the test chamber that the individual mouse spent the most amount of time in during the pretest phase. In the afternoon (4 h later), mice were given the drug and confined for 30 min to the opposite compartment. This sequence was alternated over the next 3 days. The control mice received saline in both compartments during conditioning. Drug-treated mice received saline in the morning and **KGFF09** in the afternoon. (3) In the test phase, mice were placed in the center chamber and allowed to freely move in the entire CPP apparatus. No drug or vehicle was given on this day, and the location of the mouse was measured for 15 min using EthoVision XT system (Noldus Information Technology, Wageningen, The Netherlands). Each experimental group included 8 animals.

### 3.8. Open-Field Test

The open-field test was used for evaluating the drug effect on spontaneous locomotor activity, as described previously [21]. The global locomotor activity was measured using custom-made chambers (height × width × height, 30 × 30 × 25 cm) in an isolated noise-free room illuminated to 60 Lux. On the experimental day, mice were s.c. administered saline (control) or **KGFF09**, and 30 min later were placed in the chambers, and locomotor activity was video-recorded for 30 min using the EthoVision XT system (Noldus Information Technology, Wageningen, The Netherlands). Locomotion was assessed daily for 4 consecutive days after chronic test drug or saline (control) administration, during the conditioning sessions in the CPP test described above. Each experimental group included 8 animals.

### 3.9. Data and Statistical Analysis

Experimental data were graphically processed and statistically analyzed using the GraphPad Prism Software (GraphPad Prism Software Inc., San Diego, CA, USA). In in vitro assays, potency EC_50_ (nM) and efficacy E_max_ (%) values were determined from concentration-response curves by nonlinear regression analysis. Efficacy was determined relative to the reference full MOR agonist, DAMGO. In the writhing assay, dose-response curves were constructed and the antinociceptive activity, as percentage decrease in number of writhes compared to the control group, was calculated according to the following formula: % inhibition of writhing = 100 × [(C − T)/C], where C is the mean number of writhes in saline (control) animals and T is the number of writhes in drug-treated mice. The dose necessary to produce a 50% effect (ED_50_) and 95% confidence limits (95% CL) were calculated according to the method of Litchfield and Wilcoxon [33]. In the CCP assay, data were presented as the difference in the time spent in the drug-paired compartment during the test phase and the time spent in the same compartment during the preconditioning phase. For in vivo behavioral data, two-sample comparisons were performed using unpaired *t*-test. For multiple comparisons between the treatment groups, a one-way ANOVA with Dunnett’s post-hoc test was used. All data are presented as mean ± SEM. A *p* < 0.05 was considered statistically significant.

## 4. Conclusions

The imperative need for effective and safer pain therapies in the light of the opioid epidemic continues to drive the exploration of new mechanism-based treatment strategies and the hunt for novel lead molecules. In the current study, we presented additional in vitro and in vivo pharmacological data on the dual targeting MOR agonist and NPFFRs antagonist peptidomimetic, **KGFF09**. In vitro, the earlier reported G protein-biased MOR agonist profile of **KGFF09** [19] was endorsed in our study. Using the functional assays for G protein activation, the [^35^S]GTPγS binding assay, and for β-arrestin2 recruitment, the PathHunter β-arrestin2 assay, **KGFF09** activated more favorably G protein coupling to the MOR over β-arrestin2 recruitment.

In vivo behavioral experiments established **KGFF09** to be an effective and potent antinociceptive in a visceral pain model after s.c. administration in mice, which extends and strengthens the previous pharmacological in vivo findings on the antinociceptive properties of **KGFF09** in acute pain and chronic inflammatory pain [19]. Notably, we demonstrated the absence of detrimental effects of established MOR agonists, namely rewarding effects and sedation/locomotor impairment of **KGFF09** at a dose shown to be highly effective in inhibiting pain behavior. Thus, **KGFF09** displays a favorable benefit/side effect ratio regarding these opioid-related side effects, complementing our earlier observations on reduced propensity for respiratory depression, analgesic tolerance, opioid-induced hyperalgesia and physical dependence of conventional opioid analgesics, such as morphine [19]. The dual opioid-NPFFR pharmacology of **KGFF09** within a single molecule gathers the beneficial effects of G protein-biased MOR agonism on MOR-related side effects (respiratory depression, reward and sedation/locomotor activity) and those of NPFFRs antagonism on chronic side effects (opioid-induced hyperalgesia, tolerance and withdrawal syndrome).

In conclusion, pharmacological data indicate that by activating the MOR and blocking the NPFFRs with **KGFF09**, effective antinociception effects can be achieved in a variety of pain conditions ranging from nociceptive to visceral and chronic inflammatory pain with reduced opioid-related liabilities of conventional opioids.

## Figures and Tables

**Figure 1 molecules-27-08785-f001:**
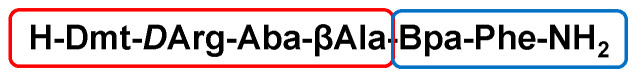
Amino acid sequence of **KGFF09**. The opioid pharmacophore is framed in red, and the neuropeptide FF pharmacophore is framed in blue. Dmt: 2′,6′-dimethyl-L-Tyr; Aba: 4-amino-tetrahydro-2-benzazepinone; Bpa: 2-amino-5-(4- benzylpiperidin-1-yl)pentanoic acid.

**Figure 2 molecules-27-08785-f002:**
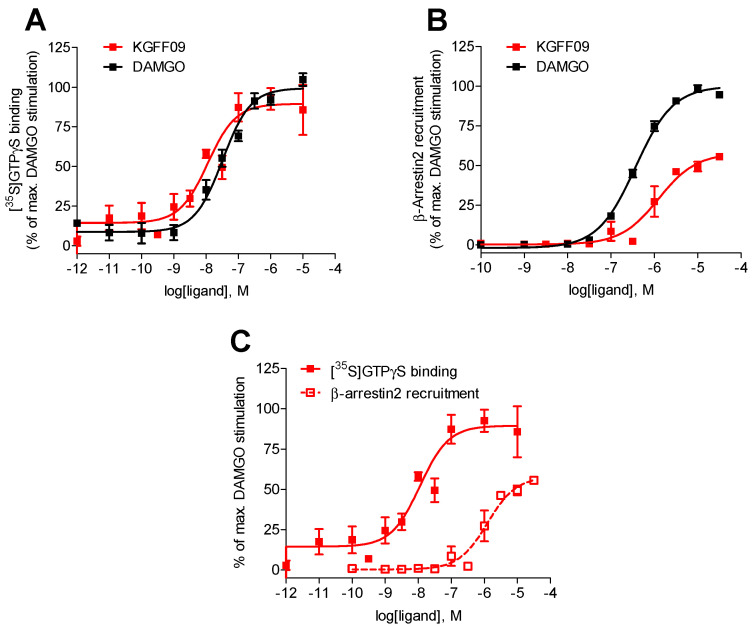
In vitro functional profile of **KGFF09** at the human MOR. Concentration-response curves of agonist-stimulated (**A**) G protein activation (the [^35^S]GTPγS binding assay) and (**B**) β-arrestin2 recruitment (the PathHunter β-arrestin2 assay) at the human MOR by **KGFF09** and DAMGO. (**C**) Comparison of G protein activation (the [^35^S]GTPγS binding assay) and β-arrestin2 recruitment (the PathHunter β-arrestin2 assay) at the human MOR by **KGFF09**. Values represent the mean ± SEM (*n* = 3).

**Figure 3 molecules-27-08785-f003:**
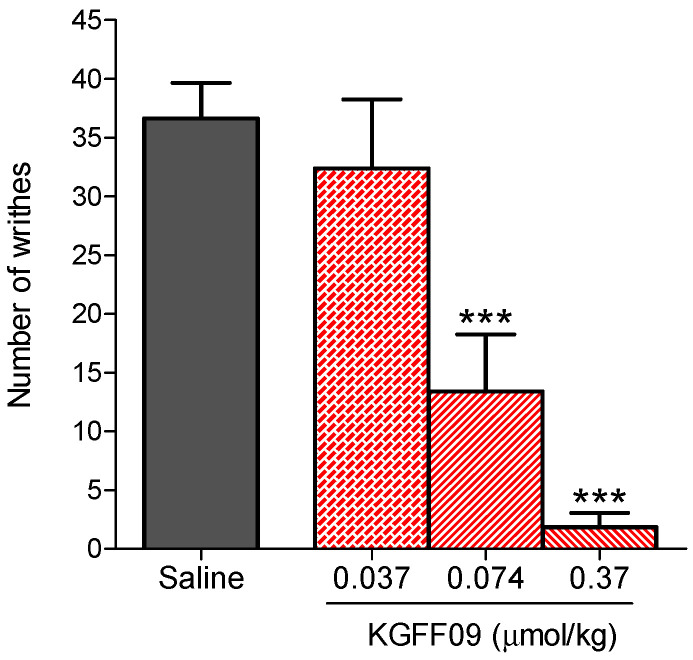
Antinociceptive effect of **KGFF09** in the acetic acid-induced writhing test after s.c. administration in mice. Groups of mice received s.c. saline (control) or different doses of **KGFF09**, and the number of writhes were counted at 30 min after drug administration for a period of 10 min. Values represent the mean ± SEM (*n* = 6–8 mice per group). *** *P* < 0.001 vs. saline group, one-way ANOVA followed by Dunnett’s post-hoc test.

**Figure 4 molecules-27-08785-f004:**
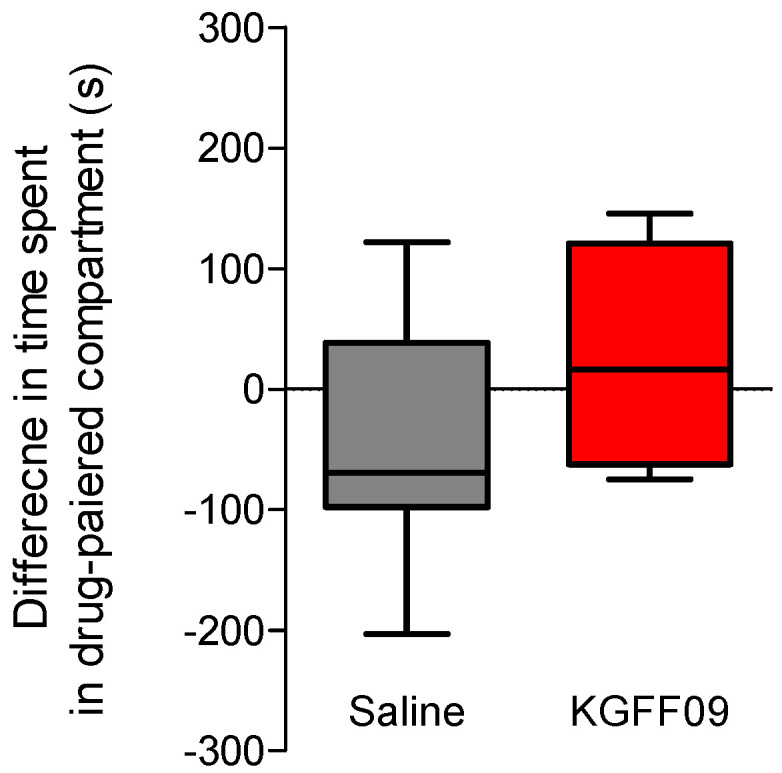
Rewarding effects of **KGFF09** in the CPP test in mice after s.c. administration. Following determination of initial preconditioning preferences, mice were place-conditioned daily for 4 days with **KGFF09** (7.4 µmol/kg) or saline. Mean differences in time spent on the drug-paired side ± SEM are presented (*n* = 8 mice per group).

**Figure 5 molecules-27-08785-f005:**
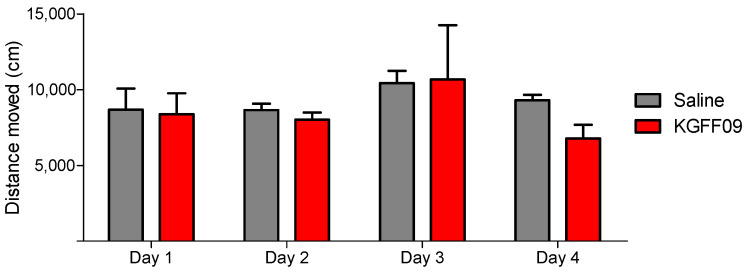
Effect of **KGFF09** on spontaneous locomotor activity after chronic s.c. administration to mice using the open-field test. Mice were treated for 4 days with **KGFF09** (7.4 µmol/kg) or saline and distance travelled in the open field was measured during 30 min. Values represent the mean ± SEM (*n* = 8 mice per group).

**Table 1 molecules-27-08785-t001:** Potencies and efficacies to the human MOR of **KGFF09** in comparison to DAMGO in functional in vitro assays for G protein activation and β-arrestin2 recruitment.

Ligand	[^35^S]GTPγS Binding ^a^		β-Arrestin2 Recruitment ^b^	
	EC_50_ (nM)	E_max_ (%) ^c^	EC_50_ (nM)	E_max_ (%) ^c^
KGFF09	8.28 ± 2.13	92 ± 9	1264 ± 388	57 ± 3
DAMGO	35.0 ± 4.2	100	367 ± 9	100

^a^ Determined in the [^35^S]GTPγS binding assay with membranes from CHO cells that stably expressed the human MOR. ^b^ Determined in the PathHunter β-arrestin2 recruitment assay with U2OS cells that co-expressed the human MOR and the enzyme acceptor tagged β-arrestin2 fusion protein. ^c^ E_max_ (%) values represent the percentage relative to the maximal effect of DAMGO (as 100%). Values represent the mean ± SEM (*n* = 3).

## Data Availability

Data is contained within the article.

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
