# Peer review of "Bifunctional Peptidomimetic G Protein-Biased Mu-Opioid Receptor Agonist and Neuropeptide FF Receptor Antagonist KGFF09 Shows Efficacy in Visceral Pain without Rewarding Effects after Subcutaneous Administration in Mice"

_molecules, 2022, doi:10.3390/molecules27248785_

Round 1

Reviewer 1 Report

This manuscript clearly demonstrated that the effect of dual ligand (KGFF09) acting as a G-protein biased MOR agonist and neuropeptide FF receptor (NPFFR) pharmacophore on the visceral pain in mice. KGFF09 showed more potent G-protein activating effect but with reduced beta-arrestin recruitment effects than those of DAMGO. Furthermore, KGFF09 exerted significant antinociceptive effect on the visceral pain induced by acetic acid. Importantly, KGFF09 did not show any rewarding effect estimated by conditioned-place preference test. From these results, authors concluded that KGFF09 or dual MOR agonists/NPFFR antagonists displays a favorable benefit/side effect ratio regarding these opioid-related side effects as improved treatments for visceral pain conditions.

The results of this study include may beneficial information to understand the therapeutic advantage of the dual ligands targeting MOR and NPFFR with reduced side effects, but I found several problems as following.

Specific comments

1. Authors had better describe the reason why they used DAMGO as the reference ligand for NPFF09, while it is not the therapeutic MOR agonist. If there are any results of the therapeutic MOR agonist such as morphine as a reference for NPFF09 especially on the visceral pain or on rewarding effects, it would be more convincing when they describe the benefit/side effect ratio regarding opioid-related side effects. 

2. In introduction or discussion section, authors had better describe or introduce more about the role of “NPFFR” on the potent antinociceptive effects or on the reduced side effects. Throughout the manuscript, it is not clear how the “dual” ligand acted on the visceral pain without rewarding effect (i.e., it is not clear whether the antagonistic effect of NPFFR is necessary for their antinociceptive effects or not. In addition, what the roles of antagonizing NPFFR of this dual ligand on the NO-induction of rewarding after repeated administration?).

3. Authors had better show the “antinociceptive effect” of KGFF09 “after repeated administration” (i.e., the same time when they showed no rewarding effect) when they describe the benefit/side effect ratio regarding opioid-related side effects. 

Author Response

Reviewer 1

This manuscript clearly demonstrated that the effect of dual ligand (KGFF09) acting as a G-protein biased MOR agonist and neuropeptide FF receptor (NPFFR) pharmacophore on the visceral pain in mice. KGFF09 showed more potent G-protein activating effect but with reduced beta-arrestin recruitment effects than those of DAMGO. Furthermore, KGFF09 exerted significant antinociceptive effect on the visceral pain induced by acetic acid. Importantly, KGFF09 did not show any rewarding effect estimated by conditioned-place preference test. From these results, authors concluded that KGFF09 or dual MOR agonists/NPFFR antagonists displays a favorable benefit/side effect ratio regarding these opioid-related side effects as improved treatments for visceral pain conditions.

The results of this study include may beneficial information to understand the therapeutic advantage of the dual ligands targeting MOR and NPFFR with reduced side effects, but I found several problems as following.

Author’s reply: We would like to thank the Reviewer for reviewing our manuscript. We also thank the Reviewer for the positive and constructive comments. We have given careful consideration to all issues raised and all changes are highlighted in the revised manuscript.

Specific comments

  1. Authors had better describe the reason why they used DAMGO as the reference ligand for NPFF09, while it is not the therapeutic MOR agonist. If there are any results of the therapeutic MOR agonist such as morphine as a reference for NPFF09 especially on the visceral pain or on rewarding effects, it would be more convincing when they describe the benefit/side effect ratio regarding opioid-related side effects.

Author’s reply: We thank the Reviewer for this comment.

We have used DAMGO in our in vitro assays because it is a well-characterized, potent and full agonist at the MOR, and a commonly employed reference MOR agonist in the [35S]GTPgS binding and β-arrestin2 recruitment assays. Furthermore, DAMGO was also used as a reference MOR agonist in our previous report [Drieu La Rochelle et al. Pain 2018, ref. 19]. We and numerous laboratories rely on the use of DAMGO as a validated, standard MOR agonist for assessing efficacy and potency of known and new ligands at the MOR in functional in vitro assays. For clarification, we have made amendments in the revised manuscript (see Results and Discussion lines 90-92 and 127-128).

We have already provided in the original submitted manuscript a comparison on the in vivo profile of KGFF09 to the gold-standard opioid morphine (see Abstract, Results and Discussion lines 154-155, 193-195 and 214-216, and Conclusions). We referred to our published data on morphine’s antinociceptive effects in visceral pain [Spetea et al. J. Med. Chem. 2019, ref 24], and rewarding effects and locomotor activity after chronic treatment in mice [Dumitrascuta et al. Molecules 2021, ref. 21]. In this current manuscript, we show that antinociceptive potency of KGFF09 in the visceral pain is 17-folds higher than potency of morphine in the writhing assay [reported in ref. 24]. Furthermore, we described recently on the significant rewarding effects (conditioned place preference test) and increased locomotor activity (open-field test) of morphine at antinociceptive doses [reported in ref. 21], whereas none of these opioid-related side effects were produced by KGFF09 in our study. All these observations establish KGFF09 as a safer antinociceptive compared to the conventional opioid morphine.

For better reading, we have made amendments in the revised manuscript (see Abstract and Conclusions).

  1. In introduction or discussion section, authors had better describe or introduce more about the role of “NPFFR” on the potent antinociceptive effects or on the reduced side effects. Throughout the manuscript, it is not clear how the “dual” ligand acted on the visceral pain without rewarding effect (i.e., it is not clear whether the antagonistic effect of NPFFR is necessary for their antinociceptive effects or not. In addition, what the roles of antagonizing NPFFR of this dual ligand on the NO-induction of rewarding after repeated administration?).

Author’s reply: We thank the Reviewer for this comment.

There is an evident separation in the pharmacology of the NPFFR system and MOR system. The NPFF1R and NPFF2R are recognized to contribute to the development of opioid-induced analgesic tolerance, dependence and opioid-induced hyperalgesia, and pharmacological blockade of these two receptors efficiently improves these adverse side effects associated with repeated exposure to opioids [Simonin et al. Proc Natl Acad Sci USA 103, 466-471, 2006 and Elhabazi et al. Br. J. Pharmacol. 165:424-435, 2012]. The MOR is established to play a major role in analgesia and the undesirable opioid side effect of reward, and deletion of the gene encoding MOR (Oprm1) completely blocks opioid analgesia and reward behavior in rodents [Matthes et al. Nature 383, 818-823].

This distinct pharmacology KGFF09 has been described in our published article Drieu La Rochelle et al. Pain 2018 [ref. 19], with KGFF09 reported as a G protein–biased MOR agonist-NPFFRs antagonist. The association of both properties within a single molecule gathers the beneficial effects of G protein-biased MOR agonism on MOR-related side effects (respiratory depression, reward, sedation/locomotor activity) and those of NPFFR antagonism on chronic side effects (opioid-induced hyperalgesia, tolerance and withdrawal syndrome), altogether leading to a potent analgesic with an improved safety profile.

We have made amendments in the revised manuscript (see Results and Discussion lines 168-171 and 174-177, Conclusions lines 362-370 and References ref. 25).

  1. Authors had better show the “antinociceptive effect” of KGFF09 “after repeated administration” (i.e., the same time when they showed no rewarding effect) when they describe the benefit/side effect ratio regarding opioid-related side effects.

Author’s reply: We thank the Reviewer for this comment.

In our published article Drieu La Rochelle et al. Pain 159, 1705-1718, 2018 [ref. 19], we have reported on the absence of antinociceptive tolerance following chronic s.c. administration (for 8 days) of KGFF09 to mice in a model of acute thermal nociception. In the same report, we stated the beneficial effects of KGFF09 on chronic side effects, including analgesic tolerance. Therefore, we did not consider necessary revaluating this behavioral response. A further important aspect to be considered is the ethical aspect of animal experimentation.

Reviewer 2 Report

The authors investigated the effects of the KGFF09 on visceral pain in mice. The results of the study show that KGFF09 provided analgesia in mice with a favorable benefit to side effects ratio. The introduction is very concise and provides sufficient background. Methods are appropriate and well described. The results are presented clearly. Appropriate methods have been used. Conclusions are supported by the results. 

Author Response

The authors investigated the effects of the KGFF09 on visceral pain in mice. The results of the study show that KGFF09 provided analgesia in mice with a favorable benefit to side effects ratio. The introduction is very concise and provides sufficient background. Methods are appropriate and well described. The results are presented clearly. Appropriate methods have been used. Conclusions are supported by the results. 

Author’s reply: We would like to thank the Reviewer for reviewing our manuscript and for the positive comments.

Reviewer 3 Report

This is a very interesting report based on a peptide KGFF09 with opioid and neuropeptide FF receptor activity. In the introduction it would be helpful to the reader if the authors detailed a little more of this peptide’s activity (line 58). It has activity at other opioid receptors (especially delta) and it antagonizes both the FF1 and 2R receptors.

Table 1/Fig 2 etc. – the authors report mean ± SEM for three or eight repetitions. SEM is a population statistic so these would be more appropriately reported as SD.

Line 145 – the dose of 0.074 µmol/kg produced an effect that was >50% of the control but your ED50 is higher than this – why is that?

Line 171 – I am not sure of the relevance of stating that this dose is 67-fold higher than the ED50 for the writhing test?

Line 199 – you have previously shown that this dose does impair motor coordination for up to 4 hours and you don’t state in your methods how long after drug administration the Open-Field test was performed.

Line 280 – does this mean that the 10-minute observation period was started 5 min after the injection of acetic acid?

Line 296 – since you only did the KGFF09 test in the afternoon is it possible that there is some circadian influence on your results?

A weakness of this study is that you only used male mice. This should be addressed in your discussion and in your conclusions i.e. the collected in vivo data only apply to male animals.

In your previous paper you stated that some authors had a related patent pending as a conflict of interest – does that no longer apply?

Author Response

This is a very interesting report based on a peptide KGFF09 with opioid and neuropeptide FF receptor activity.

Author’s reply: We would like to thank the Reviewer for reviewing our manuscript. We also thank the Reviewer for the positive and constructive comments. We have given careful consideration to all issues raised and all changes are highlighted in the revised manuscript.

In the introduction it would be helpful to the reader if the authors detailed a little more of this peptide’s activity (line 58). It has activity at other opioid receptors (especially delta) and it antagonizes both the FF1 and 2R receptors.

Author’s reply: We thank the Reviewer for this comment.

We have added two tables to the Supporting Material (Tables S1 and S2), with reference in the Introduction, summarizing the in vitro and in vivo profile of KGFF09. Although KGFF09 showed agonist activity at the delta-opioid receptor, its binding affinity is 77-fold lower than at the MOR; KGFF09 also interacts with the kappa-opioid and nociceptin receptor, but it has an antagonism profile at both receptor types. 

Furthermore, we have made amendments in the revised manuscript, i.e. Introduction (lines 168-171), Results and Discussions (lines 174-177), Conclusions (lines 366-370) and Supporting Material (Tables S1 and S2).

Table 1/Fig 2 etc. – the authors report mean ± SEM for three or eight repetitions. SEM is a population statistic so these would be more appropriately reported as SD.

Author’s reply: We do not share the opinion of the Reviewer on the unsuitable use of SEM, which is a regularly format for data presentation. We would like to retain this style.

Line 145 – the dose of 0.074 µmol/kg produced an effect that was >50% of the control but your ED50 is higher than this – why is that?

Author’s reply: We thank the Reviewer for noticing this difference, which unfortunately occurred during the preparation of the manuscript. The value of 0.11 represented the ED50 dose in mg/kg. The the correct EC50 value is 0.079 µmol/kg. Importantly, the original conclusion on the antinociceptive activity of KGFF09 has not been changed (see Results and Discussion, line 153). We apologies for this error.

Line 171 – I am not sure of the relevance of stating that this dose is 67-fold higher than the ED50 for the writhing test?

Author’s reply: We have deleted it.

Line 199 – you have previously shown that this dose does impair motor coordination for up to 4 hours and you don’t state in your methods how long after drug administration the Open-Field test was performed.

Author’s reply: The open-field test was performed 30 min after drug administration, since we found in Drieu La Rochelle et al. Pain 159, 1705-1718, 2018 [ref. 19] that the peak effect for the motor impairment was at this time point (see Materials and Methods, section 3.8).

Line 280 – does this mean that the 10-minute observation period was started 5 min after the injection of acetic acid?

Author’s reply: This is correct.

Line 296 – since you only did the KGFF09 test in the afternoon is it possible that there is some circadian influence on your results?

Author’s reply: All behavioral experiments were carried out during the light phase (between 9:00 am and 5 pm), therefore a circadian influence can be excluded. In addition, control animals were always tested in parallel to the drug-treated animals during the behavioral experiments.

A weakness of this study is that you only used male mice. This should be addressed in your discussion and in your conclusions i.e. the collected in vivo data only apply to male animals.

Author’s reply: The rational for using male mice is two-folds: (a) our earlier behavioral studies reported by Drieu La Rochelle et al. Pain 159, 1705-1718, 2018 [ref. 19] on antinociceptive effects and side effects were carried out in male mice; therefore, we have also used male mice for consistency and proper comparison of the drug effect. (b) we consider using male animals to exclude any possible influence of the menstrual cycle in female animals to the behavioral outcome due; nevertheless this is common practice in drug discovery.

In your previous paper you stated that some authors had a related patent pending as a conflict of interest – does that no longer apply?

Author’s reply: S.B. has a patent application related to hybrid mu opioid receptor and neuropeptide FF receptor binding molecules. The other authors declare no conflict of interest (see Conflict of Interest).

Round 2

Reviewer 1 Report

The authors have responded to my comments and I consider that the presentation of the work has been improved, so from my point of view the work is suitable for publication.